# Physiological Basis of Smut Infectivity in the Early Stages of Sugar Cane Colonization

**DOI:** 10.3390/jof7010044

**Published:** 2021-01-12

**Authors:** Carlos Vicente, María-Estrella Legaz, Elena Sánchez-Elordi

**Affiliations:** Team of Intercellular Communication in Plant Symbiosis, Faculty of Biology, Complutense University, 12 José Antonio Novais Av., 28040 Madrid, Spain; cvicente@bio.ucm.es (C.V.); elena_ese@hotmail.com (E.S.-E.)

**Keywords:** actin, cytoskeleton, infectivity, myosin, quorum sensing, smut, tubulin

## Abstract

Sugar cane smut (*Sporisorium scitamineum*) interactions have been traditionally considered from the plant’s point of view: How can resistant sugar cane plants defend themselves against smut disease? Resistant plants induce several defensive mechanisms that oppose fungal attacks. Herein, an overall view of *Sporisorium scitamineum’s* mechanisms of infection and the defense mechanisms of plants are presented. Quorum sensing effects and a continuous reorganization of cytoskeletal components, where actin, myosin, and microtubules are required to work together, seem to be some of the keys to a successful attack.

## 1. Introduction

Sugarcane plants are affected by multiple microorganisms, where among them is *Sporisorium scitamineum* (Syd.) M. Piepenbr., M. Stoll, and F. Oberw. (*Ustilago scitaminea* Sydow), the causal agent of smut. It currently occurs in over 64 countries and sugarcane regions, in many of which, it causes a significant amount of damage [1]. This damage increases in the sick sprouts due to secondary infections, and an increase in the size of the inoculum occurs when the whips break (a mixture of plant and fungal tissues, which is a typical structure of diseased plants) that contains spores that are spread by wind and water [2], enhancing the dispersal of the pathogen.

The plant induces several defensive mechanisms that determine the nature of the interaction with the pathogen [3]. Similarly, pathogens develop mechanisms that enable them to evade and/or suppress the defensive responses of the plant [4,5]. The entry of the infective mycelium into the vegetative bud meristem occurs within 6 to 36 h after the teliospores have been deposited onto the bud scales [6]. Active penetration of fungal hyphae from open stomata, floral organs, and even through the cuticle of the adaxial leaf epidermis have also been observed [7]. The cuticle can be mechanically destroyed as hyphae progress to the mesophilic layer of plants that are susceptible to infection [8].

The subsequent growth of hyphae within the infected plant occurs mainly in the parenchymatous cells of the lower internodes, achieving progress in its invasion by breaking the cell walls (Figure 1). However, the pathogen can also remain in a dormant phase in the apoplastic area of the parenchymatous tissue [9]. Again, hyphae development concludes with the formation of the whip (teliospore sori) in the upper internodes. 

It has been seen that different varieties of sugar cane manifest different responses to an attack. For example, the Barbados (B) 42,231 cultivar (cv.) is highly susceptible to infection. However, Mayarí (My) 55-14 is a resistant cv. that is able to defend itself against attack. Numerous studies have been directed toward explaining why sugar cane varieties respond differently to smut colonization. However, what are the triggering mechanisms of infection? Herein, we present the way in which the fungus tries to infect plants in the early stages. 

## 2. What Is the Life Cycle of the Ustilaginales-Like *Sporisorium scitamineum*?

First, to understand how the pathogen attacks, it is necessary to consider how its life cycle works. *S. scitamineum* (Syd), previously known as *Ustilago scitaminea*, is a basidiomycete belonging to the order Ustilaginales, class Ustilaginomycetes. The life cycle of the smut is simpler than that of the rust since it is developed on the same plant. In nature, the dikaryotic mycelium of these fungi seems to be the cause of its infection [10,11]. Thus, the primary mycelium is saprophytic and of short duration, with it not being infectious until it becomes dikaryotic through a process known as somatogamy.

Although there are variations in the Ustilaginales’ cycles, there are some common characteristics for the whole group (Figure 2A). The cycle includes, first, the production of teliospores within the host tissue [12]. After maturing, the teliospores, which are gathered in the sori, undergo karyogamy, by which their two haploid nuclei fuse to form a diploid [13], while acquiring a thick, dark wall that forms a blackish powdery mass reminiscent of smut.

When the conditions are adequate, the germination of the teliospores leads to the formation of the pro-mycelium that the nucleus moves toward. This will then undergo meiosis, originating four haploid cells (sporidia or basidiospores), which can divide asexually via germination, originating a variable number of sporidia. The four sporidia that are initially released, as well as the cells formed from their mitotic division, are in no case pathogenic [12].

However, the union of two of these sporidia leads to the process known as dikaryosis. Dikaryotic hyphae are infectious and involves forced parasitism in nature [10,11]. It presents a polarized growth from its extremities, which is typical of filamentous fungi [7,14]. The dikaryotic mycelium grows inside the host cells, invading the whole plant and producing the teliospores or ustilospores, which are gathered in great numbers, forming the so-called sori. These teliospores or spores of the coals are the characteristic structures of the Ustilaginals and are very important for their taxonomic classification. They are formed en masse and can develop at different places of the host, including the flowers, leaves, stems, rhizomes, and in some cases, roots.

As in the rest of Ustilaginales, the diploidal and dikaryotic mycelium of *S. scitamineum* is the one that has infective capacity since it manages to penetrate the host tissues and damage the meristematic tissues of the plant [15]. However, despite the evident proximity between *Sporisorium* sp. and *Ustilago* sp., these present certain differences in their modes of infection (Figure 2B). *Ustilago* infects the aerial parts of the plant [16], rapidly forming gills or tumors full of teliospores, while *Sporisorium* sp. infects young seedlings and can remain asymptomatic and progresses systemically [17]. In the case of this pathogen, the life cycle ends with the emergence of the whip-like structure from the shoot apical meristem, where billions of the teliospores that are produced in a single whip are easily dispersed in the field by wind, rain, and small animals [16].

At this point, it is interesting to note that infection starts again when some teliospores are deposited into other plants. As such, how does the pathogen “make sure” that a new infection will be successful?

## 3. A Quorum Signal (QS) Triggers the Infection 

To ensure an efficient attack, the size of the teliospore colony on a leaf’s surface, the buds, or a stem must reach a critical mass that puts up sufficient resistance to the possible defense mechanisms that the plant triggers in its presence. In other words, the pathogen must develop cell aggregation mechanisms such that enough infective cells survive the attack of the plant through the plant’s resistance and defense factors, i.e., “unity makes strength.” How do fungal cells communicate with each other?

Pathogenic and symbiotic bacteria depend substantially on quorum signals (QSs) to colonize and infect their hosts [18,19]. In fact, it has been seen that open stomata are initially colonized by a few bacteria, rapidly increasing the population of the pathogen over time on that structure. QSs are a system that regulates gene expression in bacteria and is dependent on the population density [20]. This enables individual bacterial cells in a local population to coordinate the expression of certain genes, helping them to behave in a similar way to a multicellular organism. QSs work via an exchange of small signal molecules between nearby bacteria. The cell population increases as a function of the signal concentration. Operationally, a bacterial quorum is present when the signal concentration reaches levels that are capable of triggering changes in the gene expression. QS is particularly important for the ability to infect the plant with pathogenic bacteria. Defective mutants in QSs are avirulent or show very reduced virulence [21].

Extracellular autoinducing QS molecules in the supernatants of microbial cultures were first recognized for their roles in the induction of genetic competence in Gram-positive bacteria [22]. However, very recently, it has been discovered that fungi can also develop QS signaling systems that affect not only the population size, but also the morphology, biofilm formation, and pathogenicity [23,24]. It seems that the common mechanism of the quorum involves the synthesis of signals that are released outside the cell via active transport or diffusion [24]. Therefore, intraspecific communication is possible because of the response to QS molecules that pathogens accumulate in their extracellular environment [22]. The natures of these QS molecules vary enormously: lactones, cyclic dipeptides, and methyl esters of fatty acids have all been described as QS inductors.

*Candida albicans*, a dimorphic fungus, was the first studied as having a QS system [22]. *C. albicans* transitions from a yeast-like growth to polarized filaments, where this capacity seems to be essential in the diffusion of the disease. Hyphae formation is spontaneously suppressed for high cell densities or when culture media are added to supernatants of cultures in the stationary growth phase. This suggested that hyphae formation is partially regulated by some soluble and diffusible factors [22]. Hornby et al. [25] identified this signal molecule as farnesol, which is active against many strains of *C. albicans* in a concentration range from 1 to 50 µM. Apart from farnesol, other QS molecules in *C. albicans* are the aromatic amino-acid-derived alcohols like tyrosol, tryptophol, and phenylethanol [26]. Other fungi have been shown to produce extracellular molecules that modulate cell morphology. *Uromyces phaseoli* produce methyl-3,4-dimethoxycinnamate, an autoinhibitor of the germination of its own spores that is effective at nanomolar concentrations [27]. In *Glomerella cingulata* cultures, a diffusible factor decreases mycelium formation with a concomitant increase in conidia formation at cell densities above 10^6^ cells mL^−1^. The chemical identity of this molecule has not yet been described [28].

Many times, QS molecules cause variations in the regulation of gene expression as a response to cell density [29] to promote invasion under optimal conditions. Interestingly, Vitale et al. [30] reveal the role of peptide pheromones in cell density regulation in *Fuxarium oxysporum* development. In a similar way, Sanchez-Elordi et al. [31] have shown that *S. scitamineum* spores in an aqueous medium secrete a glycosylated enzyme with arginase activity that binds to wall receptor points that are similar or identical to those described by Millanes et al. [32] and interact with sugarcane defense proteins. This arginase seems to have different domains of interaction with these ligands in such a way that the enzyme can adhere to one or several teliospores, producing a cytoagglutination effect. Cytoagglutination has been described as a quorum signal since the *S. scitamineum* arginase accelerates the germination of the group when it binds to the teliospore cell wall [33]. Figure 3 shows the aggregation of teliospores over time. These images were obtained after the incubation of teliospores in Lilly–Barnett medium from 0 to 72 h. During this time, the spores began to germinate and produce arginase.

Quorum signaling is such a relevant process that in recent years, many studies have been directed to elucidate the molecular mechanism that inhibits the QS in fungi in order to control pathogen development [24]. However, it is interesting that what science seeks can be found already in nature. For example, it has been described that resistant My 55-14 sugar cane plants simulate a quorum signal using their own arginase, where the production of this enzyme is strongly increased by the inoculation of healthy plants with smut sporidia. Cytoagglutination seems to be necessary to trap the teliospores in a small region of the space in contact with cane arginase, which would increase their accessibility to the cell walls and facilitate the enzymatic activities of cane glycoproteins to degrade the trapped teliospores [33]. Therefore, an efficient false quorum signal, only produced by resistant varieties, appears to be essential for the plant to simultaneously attack the maximum number of fungal cells

## 4. Pathogenicity Factors

Phytopathogenic bacteria and fungi have secretion systems through which they can inject virulence effectors into the host cells; these effectors are generally proteins but also small molecules that display varied actions on the cellular machinery. Up to the present, several of these secreted molecules have been described that, generalizing the concept, are also defined as virulence factors, given their absolute requirement to inject into the plant to develop disease [34]. 

*S. scitamineum* possess a diverse range of effectors that are directed toward manipulating the host metabolism [21,35,36] and to defending itself from the plant’s immune system. From active-growing mycelium *S. scitamineum*, a group of proteins that develop a biological action on the host tissues has been isolated via extraction in aqueous medium and penetrability chromatography [37]. The most active fraction (fraction 5) only contained protein, while the rest of the fractions, of which the most active were 6 and 7, were glycoproteins. After incubating the leaf discs of the smut-sensitive Barbados (B) 42231 cv. and the resistant Mayarí My 55-14 cv. in solutions of these fractions, it was observed that the total phenol content was similar in the control discs of untreated and treated plants at time zero, but it increased notably after the incubation of the discs with the protein of fraction 5, and to a lesser extent, although very significantly, with the glycoproteins contained in fractions 6 and 7. The varietal difference implies a greater accumulation of free phenols in the leaves of the sensitive cultivar. This increase somehow corresponds to the activation of the phenylalanine ammonia lyase (PAL) enzyme, which is higher in My 55-14 than in B 42231, as well as the peroxidase (POX) system. The lower PAL activity in the sensitive cultivar could be related to the levels of caffeic acid and its derivative, chlorogenic acid, which remain high in cv. B 42231 but not in cv. My 55-14. The accumulation of caffeic acid will then produce a feedback inhibition of the PAL such that its detectable activity levels in the sensitive varieties would be lower. This would indicate that virulence factors, i.e., both proteins and glycoproteins, would activate phytoalexin synthesis in the sensitive cultivar (phenols derived from hydroxycinnamic acids, such as flavonoids or tannins), while in the resistant cultivar, phenols derived from a PAL activity would be preferentially directed toward lignin and lignan synthesis, in which peroxidases play an essential role (Figure 4). In this sense, the production of lignans has been considered a defense factor of the plant, in addition to the production and secretion of defense proteins [38]. In fact, the way in which sugar cane plants direct their resources to lignin/lignan synthesis by means of the regulation of dirigent protein levels seems to be critical in the early defense of resistant My 55-14 cvs. [38,39].

In vitro studies evidence that some proteins related to cell wall modification, morphogenesis, polysaccharide degradation, and carbohydrate metabolism are exclusively secreted in response to host extract media, probably at early time points during the penetration and colonization of sugarcane cells [40]. Recently, Teixeira-Silva et al. [41] have detected *S. scitamineum* effectors, which are critical molecules that are be able to defeat sugar cane defense mechanisms. Various potential interactors were identified, including subunits of the protein phosphatase 2A and an endochitinase. The results evidence that the expression of effectors is influenced by the sugarcane genotypes during tissue colonization. Moreover, it seems that orthologs of sugarcane that share around 70% similarity may be the plant targets of these effectors.

Some species of corn smut, such as *U. maydis*, have genes coding for virulence factors that are secreted into the environment. Interestingly, the *S. scitamineum* sequenced genome contains more genetic clusters or groups for these secreted effectors than other similar species [36]. This increase in the number of genes appears to be due to tandem genetic duplication and to the existence of other elements that are associated with these genes [42]. Some of these genes are related to the synthesis of enzymes that are responsible for the degradation of the cell wall, and because of that, they are clearly involved in the development of the virulence of the pathogen [36]. 

During the coevolution of fungal plant pathogens and their hosts, a seesawing interplay between pathogen virulence and host resistance has been developed. It is an interesting point of plant–pathogen coevolution that the early defense of sugar cane is also focused on cell wall degradation. Enzymes, such as chitinases [43], glucanases, and peroxidases, and sometimes catalases [44], have been traditionally related to the defense mechanisms of the plant. In the same way, polyamine accumulation in *S. scitamineum*, at moderate levels, is considered a virulence factor that is necessary for smut growth and pathogenicity [45] as in other Ustilaginales [46]. In *U. maydis*, putrescine is an essential molecule for the transformation into infective mycelium, as sporidia that are unable to produce putrescine cannot achieve the transition to the dimorphic state [47]. The formation and growth of dikaryotic hyphae after sexual mating is critical for *S. scitamineum’s* pathogenicity; Chang et al. [48] demonstrated that an elevated intracellular ROS (reactive oxygen species) level promotes *S. scitamineum* mating filamentation via the transcriptional regulation of ROS catabolic enzymes and is regulated by the cAMP/PKA signaling pathway. A connection between putrescine production and ROS has been suggested [49,50], where moderated levels of that polyamine may be involved in sporidia transformation into dikaryotic mycelium.

However, increased production of these molecules induced by sugarcane glycoproteins leads to nuclear decondensation, cell wall breakdown, and germinative blockage of teliospores as a part of the plant’s defense mechanism [45]. It seems that the germination blocking by means of a polyamine level increase occurs through the disruption of the cytoskeleton. Moreover, the role of sugar cane arginase (the one that is responsible for false quorum signals) in cytoskeletal disorganization has been amply studied [50,51]; as such, how important is an organized cytoskeleton in teliospores for pathogenicity and development? 

## 5. The Role of the Cytoskeleton in Teliospore Germination

Fuchs et al. [52] showed that F-actin is essential for polarized growth during the infection of corn with *U. maydis*, a pathogen that is phylogenetically related to *S. scitamineum.* In addition, through their role in cytoskeleton organization, Rho GTPases are required for the establishment and maintenance of cell polarity, as well as polarized growth in fungal cells [53]. Numerous experiments in the presence of the inhibiting agents of cytoskeletal functionality have demonstrated the involvement of actin in the establishment of cell polarity. Incubation with latrunculin A, a depolymerizing agent of actin filaments [54], leads to depolymerization of the filaments in *Saccharomyces cerevisae* cells, compromising their sporulation [55].

The main cellular event that precedes the germination of smut teliospores is the establishment of cell polarity [56]. These results are in agreement with those obtained by Bachewich and Heath [57], who demonstrated that F-actin participates in such polarization and in the beginning of the protrusion of the end of a nascent hypha in *Saprolegnia ferax*. Millanes et al. [58] found a total absence of cell polarization in non-germinated, resting teliospores, as revealed by the absence of staining of F-actin with fluorescein isothiocyanate (FITC)-labeled phalloidin (Figure 5A), but G-actin molecules began to polymerize and polarize after teliospore incubation in Lilly–Barnett medium (Figure 5B). After staining with FITC-phalloidin, actin capping marked the site on the teliospore wall through which the germ tube emerged (Figure 5C).

The cytoplasm polarization marked by the formation of F-actin microfibrils is prevented by the use of several inhibitors, such as phalloidin, an actin cytoskeleton stabilizer that prevents the depolymerization of F-actin [59], latrunculin A, which is a chemical agent that keeps actin depolymerized [54], or blebbistatin, which is a myosin II contractility inhibitor [60]. In addition, several glycoproteins produced by the host plant, particularly that which exhibits arginase activity, can inhibit actin capping and, therefore, prevent teliospore germination.

The organization of the actin in the early stages of the pathogen’s life cycle (Figure 6B) confirmed that an organized distribution of F-actin should be a key condition for the proper progress of sporidia emergence [51]. In this contribution, information gathered from the micrographs obtained by confocal microscopy made it possible to draw up a scheme of the organization of the microfilaments during the early stages of development. F-actin and myosin were jointly located in situ in the teliospores during the first stages of their germination, as can also be seen in Figure 6A. It was found that actin and myosin are distributed throughout the cellular cytoplasm (including the cortical zone in the case of actin) in the teliospores. Both proteins co-localize inside the cell, which suggests that they act together. After the emergence of hyphae, it was observed that actin marking tended to disappear in the teliospore (mainly in the cortical region), intensifying in the cytoplasm of the germinative tube, whose growth was then active [51]. Thus, the teliospores transport new material in the direction of the end of the nascent hyphae, as occurs commonly in other filamentous fungi [14]. However, while the cellular content is translocated to the interior of the emerging germinative tube, the teliospore loses its function and degenerates, thus gradually diluting the F-actin mark.

Microtubules (MTs), as actin filaments, are also involved in *S. scitamineum* germination [50]. While latrunculin (an inhibitor of actin polymerization [54]) blocks sporidia formation and therefore does not allow germination to begin, nocodazole (a drug responsible for MTs disorganization [61]) completely prevents sporidia release but not germinative tube formation. In this case, germinative tubes are formed but they cannot be released until the nucleus reaches the body of the hyphae; sporidia liberation must “wait” for nucleus repositioning. Therefore, MTs in *S. scitamineum* are not necessary for hyphae elongation, but they may be involved in the nuclear repositioning of the growing hyphae. Such mechanistic separation has been observed in other systems [62,63,64].

The presence of nuclear migration with the help of the cytoskeleton during germination is supported by other experiments. The use of the fluorescent antibodies anti-α-actin and anti-tubulin has allowed for obtaining micrographs using a confocal microscope that clearly indicates the simultaneous presence of both polymeric systems, namely, microfibrils and MTs (Figure 7, series 1 and 3), which were both in the nuclei of the teliospores and were revealed by means of staining with 4.6-diamino-2-phenylindol (DAPI), and in the cytoplasm of the hypha in active growth (Figure 7, series 2 and 3). The coexistence of both systems, namely, microfibrils and microtubules, clearly indicates that the nucleus can move in a directed way through the actively growing hyphae once the teliospore has germinated, as is presented in previous works [50].

## 6. The Role of the Cytoskeleton in Teliospore Motility

Because the pathogen can use the open stomata and other openings of the host as a route of entry to penetrate the internal tissues [7], it is thought that teliospores deposited at random on a leaf’s surface should develop mechanisms of displacement toward the entry pathways to the plant. Brand and Gow [65] summarize the current knowledge regarding spore movement during plant–pathogen interaction. The two most frequently suggested mechanisms are submicroscopic contractions of helically distributed fibrils in cell walls and the existence of mobile appendages in zoospores. Other species of pathogenic fungi produce spores that can move via a mechanism called gliding. This type of movement differs from dragging or swimming in that gliding does not involve the action of any apparent external motility or any obvious change in cell size, and furthermore, always requires the presence of a substrate. The cytoskeleton of *S. scitamineum* cells is also involved in the movement of cells to a chemotherapeutic agent produced by the host plant [66]. Numerous models of chemotaxis have been described regarding plant–pathogen interactions, where motility is an important feature of virulence [67].

Chemotaxis in eukaryotes involves the differentiation of two cellular poles during migration: one directed toward the source of the chemoattractant, in favor of the gradient, and the other in the opposite direction. These poles show differences in the distribution of their components, specifically those involved in the signaling and rearrangement of cytoskeleton proteins in plants and animals [68,69]. Thus, during chemotaxis, the establishment of a cellular asymmetry constitutes the basis of polarized movement. 

Experiments carried out by Sánchez-Elordi et al. [66] evaluated the migration of teliospores toward sugarcane glycoproteins. Curiously, migration was more efficient after contact with glycoproteins from resistant plants, which suggests that induced chemoattraction is an early resistance mechanism that is probably directed to enhance the false quorum signal [33,50,51]. On the other hand, the displacement of *S. scitamineum* teliospores was radically diminished in the presence of inhibitors of cytoskeleton organization (latrunculin A, phalloidin, and blebbistatin), which confirmed that the cytoskeleton must be directly involved in the motility of the teliospores toward the chemo-attracting agent. Micrographs obtained using SEM and TEM (Figure 8) demonstrate the absence of external structures related to motility in teliospores that were previously exposed to the chemoattractant (Figure 8A,B). However, the presence of invaginations, which were clearly visible, in one of the cellular poles stands out (Figure 8C). It has been hypothesized that these invaginations arose because of cytoskeletal reorganization to direct the cell toward the chemoattractant in a liquid medium or, at least, on solid surfaces (leaves, stalks) covered by drops of condensed water [66].

## 7. The GTPases as Mediators of the Signal of Organization of the Cytoskeleton

When circumstances become favorable, processes of chemoattraction, cytoagglutination by QS molecules, and germination are induced in the *S. scitamineum* population. Therefore, the cytoskeletal reorganization of teliospores is the result of a specific and controlled response to environmental conditions. GTPases have traditionally been described as the link between the reception of an external signal and its transduction inside the cell to produce a physiological response [70]. Thus, the Rho GTPases act like switches of large signaling cascades by alternating their active and inactive states, binding to GTP or GDP, respectively [71]. The signals are often derived from extracellular ligands that activate guanosine nucleotide exchange factors (GEFs) that catalyze the replacement of GDP by GTP in the GTPases, allowing their activation. Only in their active conformation, when bound to GTP, can Rho proteins interact with specific effectors inside the cell [72]. Among all the possible intracellular responses, their activity directly influences the organization of the cytoskeleton [73]. 

The organization of the cytoskeleton in *S. scitamineum* cells is also a consequence of a signaling cascade triggered by the action of GTPase proteins [66]. The addition of GTP to the teliospores’ incubation medium partially reverses the inhibitory effect of latrunculin A, probably through Rho activation; in contrast, neither the addition of GTPγS, a non-hydrolyzable analog of GTP that keeps GTPase activated [74], nor the addition of GDPβS, a deactivator of Rho GTPases [75], were able to reverse the inhibitory effect. These results indicate that GTPγS may behave as a strong Rho activator such that cells may respond to this hyperactivation by inactivating the pathway. This suggests that Rho proteins should play an important role as a communication axis in the production and modulation of cellular responses to different forms of stress [76]. On the other hand, GDPβS blocks the signaling cascade, possibly by inhibiting the GDP–GTP exchange in Rho. As a result, in both cases, the transduction of the cascade remains blocked in the cells.

Many GEFs are catalytically inactive when they form myosin complexes [77]. It has been described that blebbistatin can exert its inhibitory action on myosin functionality through the release of GEFs that stimulate the activation of Rac GTPase proteins in many cell types [77,78]. It is thus suggested that, in the presence of blebbistatin, the Rac-GTPase-protein-directed pathway should be active, which triggers the inhibition of the protein kinase that is responsible for the phosphorylation (and subsequent activation) of myosin [71,79]. Therefore, a study was conducted to determine whether GTP and its analogs caused the reversal of the inhibitory effect induced by blebbistatin. Because blebbistatin can induce disruption of actin–myosin interactions, the integrity of F-actin was assured in the presence of phalloidin. The addition of GTP to the blebbistatin-containing incubation medium caused even greater inhibition of teliospore motility than the drug alone, probably by stimulating the pathway that led to myosin dephosphorylation. However, the presence of GDPβS, GTPγS, and above all, the joint addition of GTPγS and GTP succeeded in reversing the inhibitory effect of the drug. The GDPβS blocked the route that inhibits the functionality of myosin, so it must be active in the presence of the analog [66].

The recovery of mobility in the presence of GTPγS could again be the consequence of the inactivation of the route by hyperactivation of the Rac GTPase protein, which would avoid the blocking of the kinase that phosphorylates (and activates) myosin. Since the actions of the Rho and Rac GTPases are usually antagonistic [71], the inactivation of Rac would in turn lead to the activation of the Rho-triggered pathway (Figure 9). Therefore, cell displacement is further stimulated during simultaneous incubation in the presence of GTPγS and GTP.

During cell migration, the formation of protrusions as a result of the reorganization of actin filaments via the action of myosin II has been described in different eukaryotic cell types, including *Dictyostelium*, leukocytes, or fibroblasts [80,81]. The formation of filopodia and lamellipodia in all these cell types implies the translocation of the filaments toward the back of the protrusion as a result of the retrograde flow of the actin to create free space in the forward pole that allows for its polymerization in the positive (+) end of the filament [82]. Based on what occurs during cell displacement in other organisms, a movement model based on the reorganization of F-actin, in collaboration with myosin II (no protrusion formation), has been described for the migration of *S. scitamineum* cells [66] and is schematized in Figure 9.

First, the translocation movement of the actin filaments toward their negative end would produce the invagination of one of the cell poles. In turn, the translocation must be originated by the pushing movement of myosin II, which displaces the microfilaments toward the equatorial axis of the cell (negative (−) end of the filament). As a sine qua non condition, the actin filaments must be attached to the plasma membrane at one of the cell poles using membrane anchor proteins. In this way, when the filaments are “pushed” into the cell, they “pull” the plasma membrane, producing invagination at one of the poles [83]. Once the invagination is produced, the spore must move forward (Figure 9). As the polymerization is not interrupted, the positive (+) end located at the non-invaginated end grows rapidly; therefore, it ends up “pushing” the cell membrane, undoing the invagination produced at the back end, favoring the advance of the teliospore, and allowing the cycle to begin again.

On the other hand, MTs are also represented in the scheme of Figure 10 showing why they participate in teliospore displacement. It was observed that MTs were involved in the motility of teliospores since nocodazole (Noc) decreased its displacement to a maximum concentration of 0.5 µg mL^−1^ [66]. Moreover, Figure 11 shows variations in the agglutination of the spores in the presence of Noc. The results are expressed as a percentage of cells involved in each of the different groups. It can be observed that the size of the cytoagglutinations formed in the presence of the drug decreased, probably as a result of the inhibitory effect of Noc on chemotaxis [66]. Specifically, in the presence of Noc, it was observed that a greater number of cells involved in the medium-sized groups (51 to 200 spores), which was accompanied by the absence of large clusters (groups of 200 to 500 teliospores).

On the other hand, it was proven that the dynamic interactions between the MTs and actin filaments direct the migration process in fibroblasts, where the complete depolymerization of the MTs totally blocks its displacement [84]. The interaction between MTs and actin filaments is a basic phenomenon that underlies all cellular processes in which it is necessary to establish and maintain cellular asymmetry [85], with these being some of the few cellular types in which it has been found that the cellular movement is totally independent of the MTs [84]. The interaction of actin and MTs in *S. scitamineum* cells was found. Labelling experiments of F-actin with phalloidin conjugated with Alexa Fluor^®^ 488 showed differences in the organization of actin filaments in the absence and in the presence of Noc at a concentration of 0.5 µg mL^−1^. The decrease in the number of cells showing an asymmetric arrangement of F-actin inside the cell in the presence of the inhibitor confirmed an interaction between the microfilaments and MTs during the establishment of cell polarity in *S. scitamineum* [50].

Control cells, which were not treated with any of the described inhibitors, revealed a homogeneous distribution of actin filaments by the cytoplasm before cell polarization, while the MT showed a directed polymerization (polarized) toward a single point of the cell, probably the one corresponding to the germinative pole, which was co-indicated with the actin capping when the cell polarized [50].

Finally, although cytoskeleton organization during chemotaxis and germination is essential during *S. scitamineum’s* early pathogenicity, it is important to point out that MTs could be involved in the conjugation process by means of their interaction with the actin cytoskeleton [52]. In addition, MTs are traditionally required to generate cytokinetic phragmoplast during cell division in plants [86]; therefore, polymerization could also be involved in hyphae septation, which would explain why germinative tubes (even if empty) are not released to the media in the presence of nocodazole [50].

## 8. Conclusions

It can be concluded that some cellular processes are essential in *S. scitamineum* during sugar cane infection. First, arginase is synthesized by teliospores of *Sporisorium scitamineum* when environmental conditions are adequate for their proliferation. Synthesized and secreted arginase induces a QS effect that promotes infection since it agglutinates enough teliospores. Moreover, the arginase produced in the early stages stimulates germination of the cytoagglutinated teliospores that were “ready for the attack.” On the other hand, the formation of protrusions that were derived from cytoskeleton interactions is indispensable for cell migration, namely, with actin, myosin, and MTs, as well as the signaling cascade mediated by the GTPases that collaborate in both germination and displacement

Finally, it is easy to understand why plant resistance mechanisms are focused on trying to reorganize the teliospore cytoskeleton “at its whim.” This is why chemoattractive proteins trigger cytoskeleton organization to collect teliospores to a cytoagglutination point. However, at the same time, sugar cane arginase causes cytoskeleton disorganization, where germination does not take place and infection does not progress.

## Figures and Tables

**Figure 1 jof-07-00044-f001:**
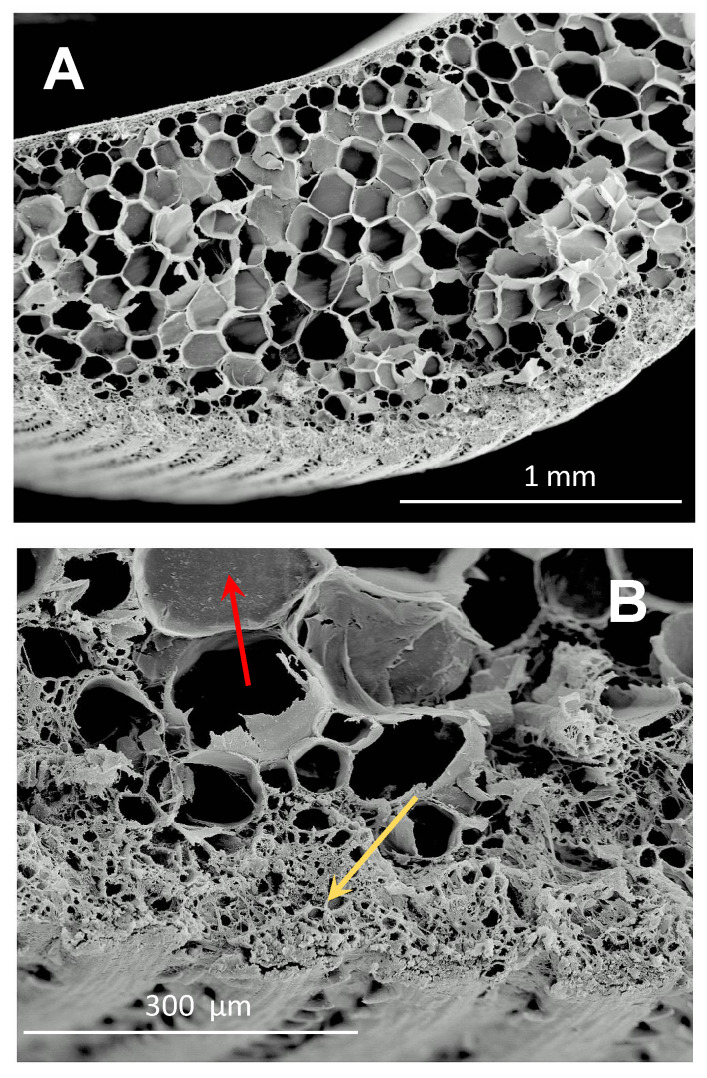
(**A**) SEM micrograph of a cross-section of a healthy sugar cane leaf of cv. Louisiana 55-5. (**B**) SEM micrograph of a cross-section of a smut-diseased leaf of sugar cane cv. Louisiana 55-5. In this cut, it is possible to observe some obturated xylem vessels (red arrow) and the subepidermal and parenchymal tissue that was invaded by the fungal mycelium (yellow arrow).

**Figure 2 jof-07-00044-f002:**
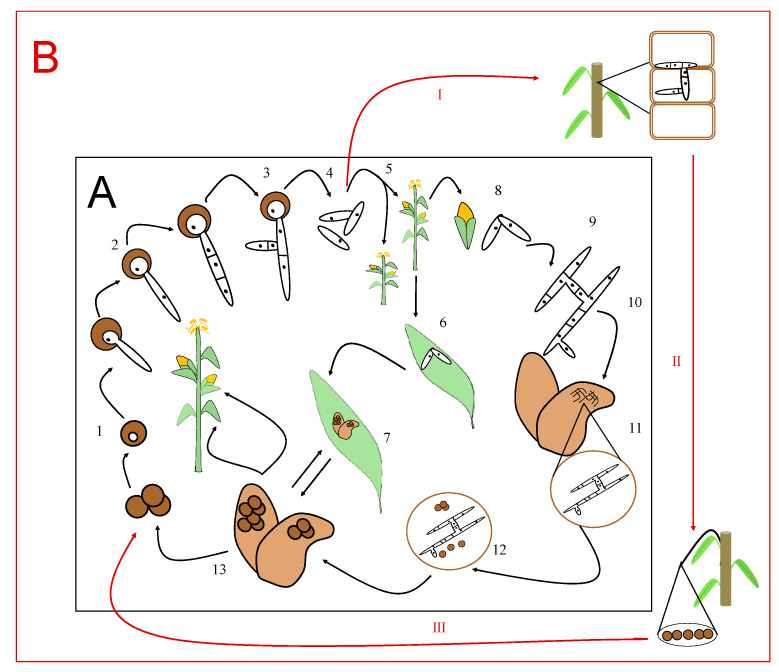
(**A**) Life cycle of *Ustilago maydis* with the stages represented as follows: (1) teliospores overwintering on soil, (2) germinating teliospore, (3) basidium, (4) basidiospores, (5) infection by the basidiospores of young plants or growing tissues in older plants, (6) leaf infection by compatible basidiospores, (7) galls on leaves, (8) corn ears infection by compatible basidiospores, (9) dikaryotic mycelium formation, (10) mycelium enlarges and forms galls in corn kernels, (11) mycelium in galls, (12) dikaryotic cells of mycelium become teliospores in galls, and (13) galls full of teliospores. (**B**) Differences in the life cycle of *Sporisorium* sp. as represented by (I) sugar cane meristematic infection by basidiospores, (II) emergence of the whip-like structure from the shoot apical meristem, and (III) billions of teliospores produced in a single whip.

**Figure 3 jof-07-00044-f003:**
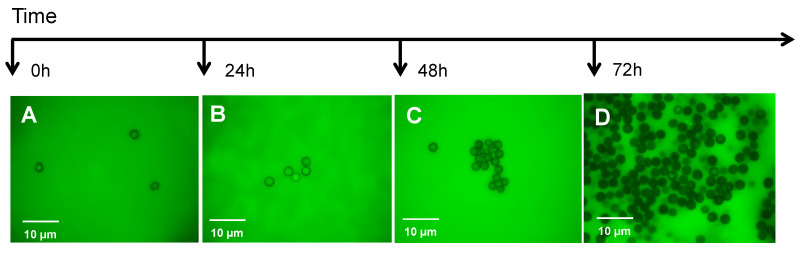
Quorum sensing of smut teliospores via the action of its own arginase that was secreted into the Lilly–Barnett medium. The degree of cytoagglutination was displayed after 0 h (**A**), 24 h (**B**), 48 h (**C**), and 72 h (**D**) of incubation.

**Figure 4 jof-07-00044-f004:**
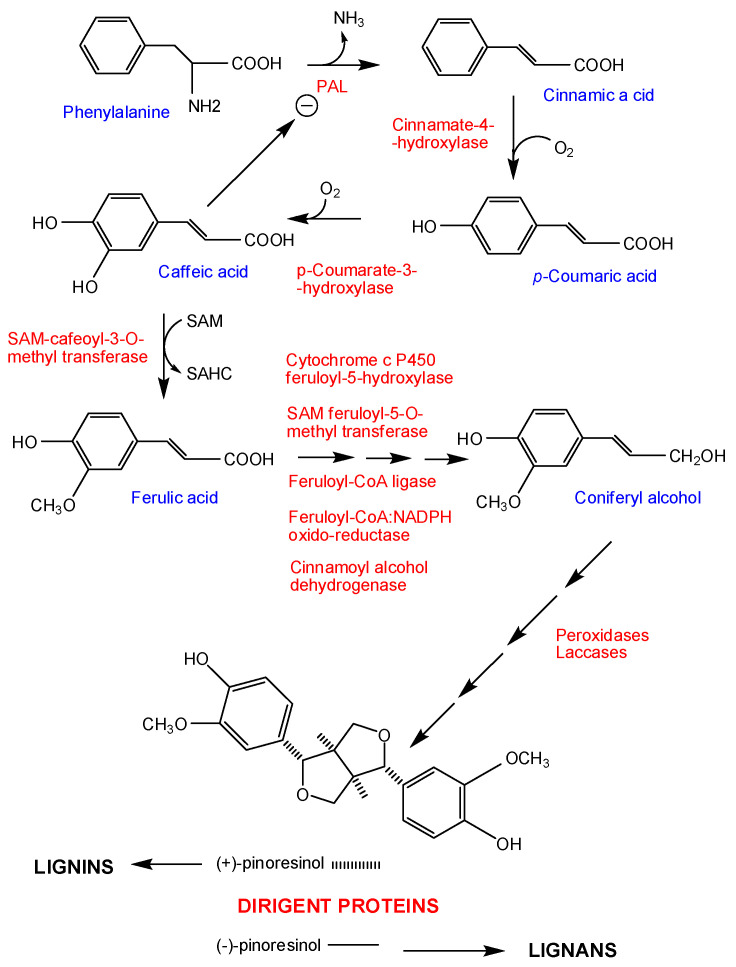
Caffeic acid acts as a regulator of the initiation of lignin biosynthesis via the feedback inhibition of phenylalanine ammonia lyase in smut-sensitive cultivars.

**Figure 5 jof-07-00044-f005:**
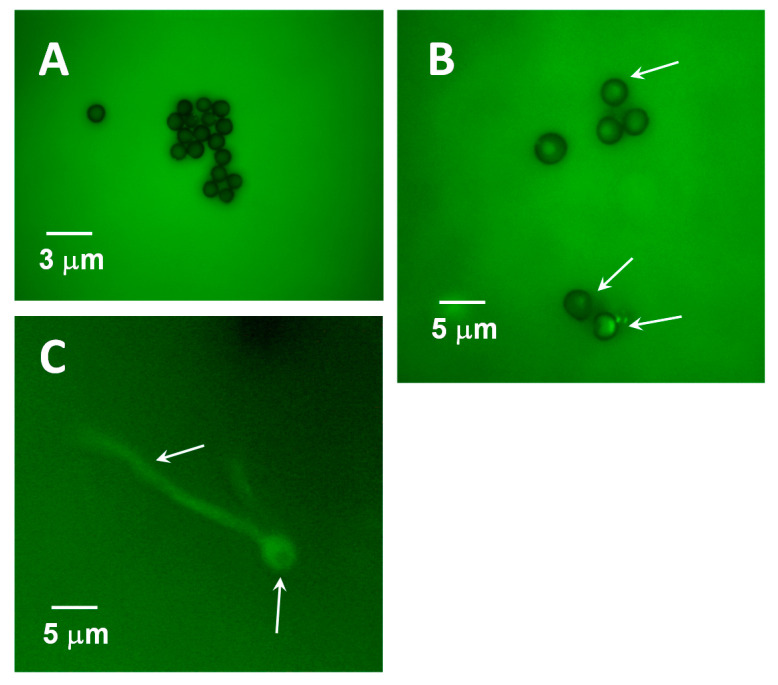
Polarization of the F-actin microfibrils and appearance of capping after the incubation of teliospores with fluorescein isothiocyanate (FITC)-phalloidin. (**A**) Resting cells in which capping did not appear. (**B**) Initiation of capping (arrows) after a few hours of rehydration. (**C**) A germinated spore, with the hypha emerging from the teliospore through the germinative pore, as signalized by the capping. The dark zone opposite to this germination point can be observed (arrow).

**Figure 6 jof-07-00044-f006:**
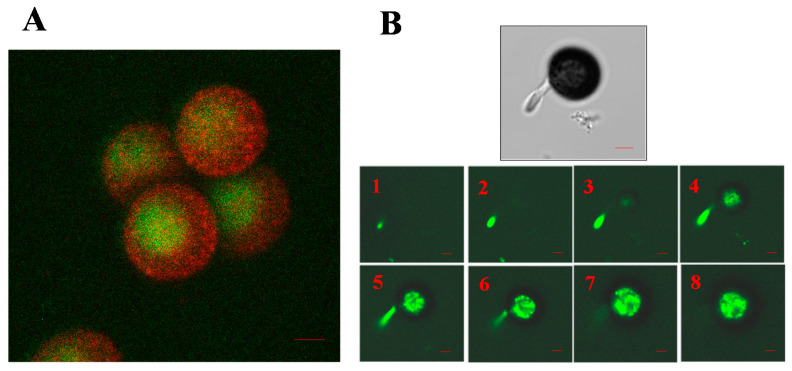
(**A**) Image obtained using confocal microscopy that displays to a group of teliospores. The green color corresponds to Alexa-Fluor^®^-488-conjugated phalloidin labeling. The red color corresponds to labeling with an anti-phosphorylated MLC-Ser19 antibody. Scale bars indicate 2 µm. (**B**) Series of images obtained using confocal microscopy that displays a germinating teliospore labeled with Alexa-Fluor^®^-488-conjugated phalloidin. Scale bars indicate 2 µm.

**Figure 7 jof-07-00044-f007:**
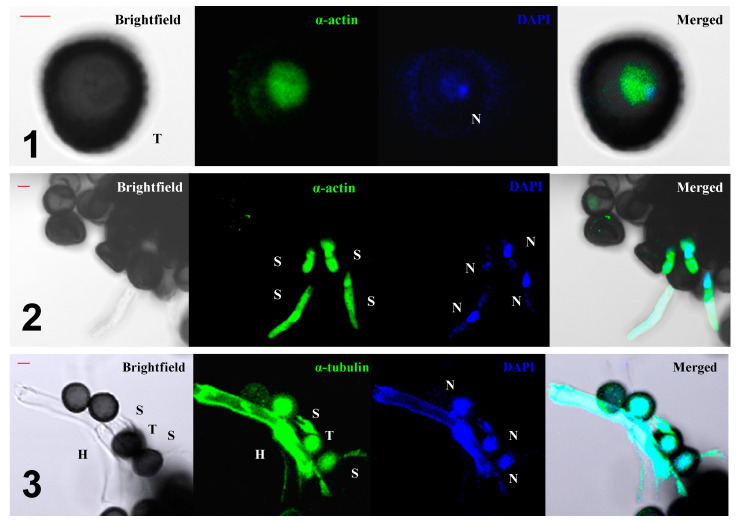
Confocal series (z-maxprojection) of cells. Brightfield, DAPI, α-actin images, as well as brightfield/DAPI/α-actin composites, are shown in the 1 and 2 series. Brightfield, DAPI, α-tubulin images, as well as brightfield/DAPI/α-tubulin composites, are shown in series number 3 (N—nucleus; H—hyphae; S—sporidia; T—teliospore). Scale bars indicate 2 µm.

**Figure 8 jof-07-00044-f008:**
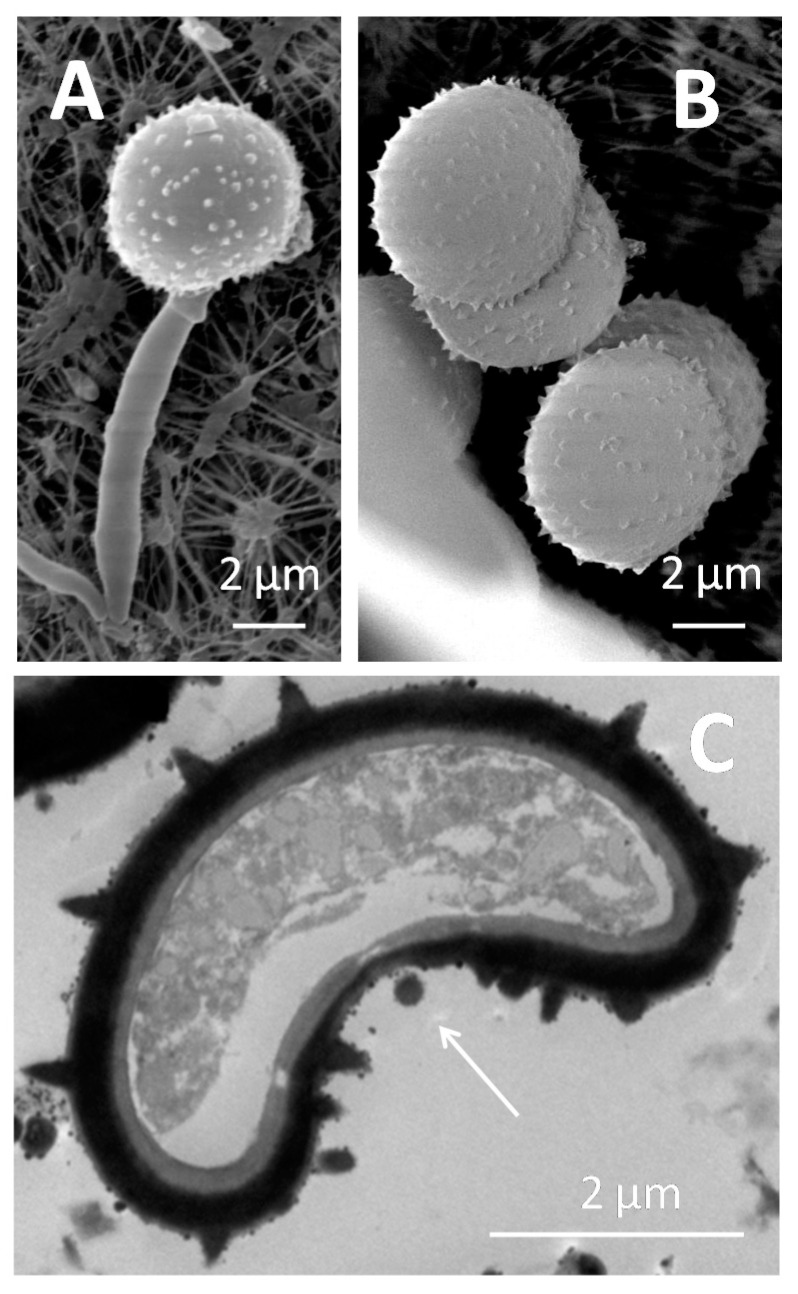
SEM micrographs of (**A**) a germinated teliospore and (**B**) aggregated teliospores after 5 h of chemotactic stimulation with sugar cane glycoproteins from the resistant My 55-14 cv. showing superficial, non-motile appendages. (**C**) TEM micrograph of a teliospore that was chemotactically stimulated with sugar cane glycoproteins from the resistant My 55-14 cv., where the white arrow indicates the invaginated pole opposite the emergence point of the nascent hypha. This zone, opposite to the invaginated pole, would mark the advance front of the teliospore in a liquid medium.

**Figure 9 jof-07-00044-f009:**
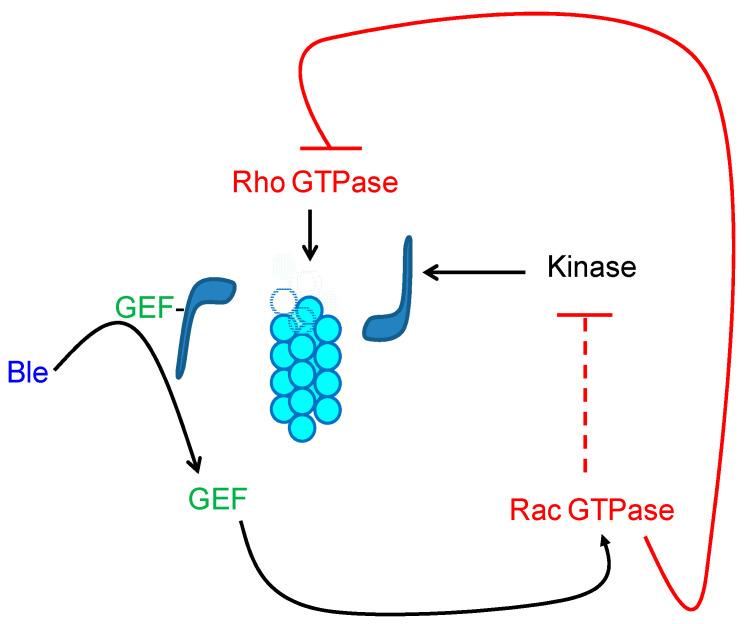
Diagram representing how the presence of blebbistatin (Ble) could trigger the activation of the Rac GTPase by means of the release of their guanosine nucleotide exchange factors (GEFs). The activity of the Rac GTPase would block myosin phosphorylation via the corresponding kinase. In turn, the Rac GTPase inhibition would stimulate Rho GTPase activation and polymerization of the actin. 
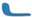
: myosin II, 
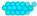
: actin filaments.

**Figure 10 jof-07-00044-f010:**
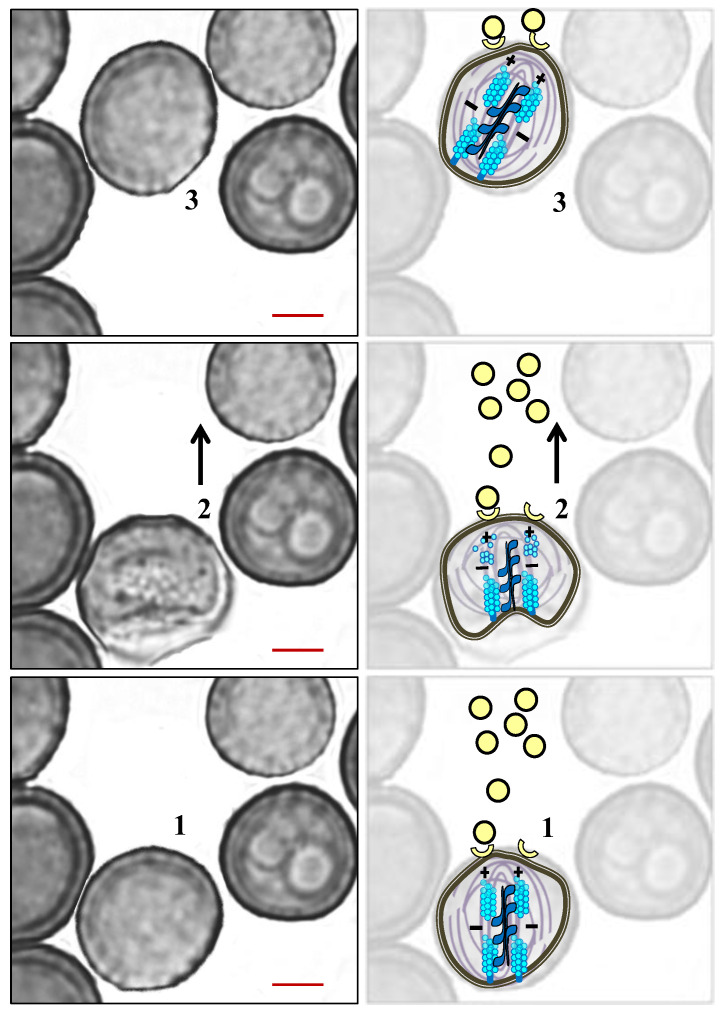
Schematic representation (on the right) superimposed on a real image (on the left) that demonstrates the role of the cytoskeleton in teliospore motility. The binding of the quorum signal to its receptors (1) induces polar cell invagination (2), which is produced by the interaction of an ATPase with contractile ability, which is sensitive to blebbistatin, with F-actin cytoskeleton. After this, the depolymerization of F-actin is achieved at the opposite pole, the repolymerization of which produces the cell advancement (3). Throughout this process, microtubules (MTs) are reorganized in cells to collaborate in the displacement with F-actin filaments. Scale bars indicate 2 µm.

**Figure 11 jof-07-00044-f011:**
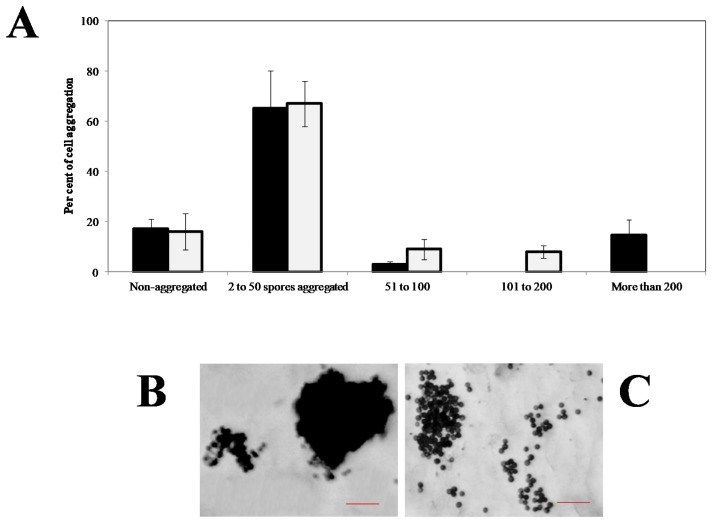
Cytoagglutination degree (number of teliospores per aggregate and number of total aggregates in the media) that was induced by distilled water (black) or 0.5 µg mL^−1^ nocodazol (gray). Values are the mean of three replicates. Vertical bars give the standard errors, where larger than the line (**A**). Teliospores aggregate formation induced by water (**B**) or nocodazol (**C**). Scale bars indicate 50 µm.

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
