# Peer review of "Physiological Basis of Smut Infectivity in the Early Stages of Sugar Cane Colonization"

_jof, 2021, doi:10.3390/jof7010044_

Round 1
Reviewer 1 Report
In this work, Sánchez-Elordi et al. present a review on selected aspects of the infection mechanisms used by Sporisorium scitamineum to infect sugar cane. Most of the work in this pathosystem have been done from the perspective of the plant. Thus, summarize and put in perspective the knowledge on this important plant pathogen is of interest. However, the review falls short in that it does not really cover the full knowledge on pathogenic mechanisms for this fungus and neither they are immersed in the general knowledge of other plant pathogenic fungi. Due to the reduced knowledge on this fungus, I consider important to cover the whole data available and compare it with the knowledge on other pathogenic fungi in a more systematic way. Authors focus on two mechanisms, quorum sensing and cytoskeletal component rearrangements. Although these two aspects of infection are of interest, general effector biology is currently a field of great interest and several genomic, transcriptomic and proteomic works available for S. scitamineum highlight the presence of virulence factors with effect on different biology process (Que et al. 2014; Su et al. 2015; Dutheil et al. 2016; Barnabas et al. 2017). In addition, the filamentation and mating processes are essential process for infection and several works describing regulators of this processes exists in this fungus. Thus, I consider that the systematic review of the infection mechanisms present in S. scitamineum rather than focusing in just two selected ones would be of a better interest.
In addition, I have other general comments that I list below:
- A general problem I have found along the manuscript is the lack of references in important parts of the text. Frequently, this lack of references make difficult the review process, and it will make difficult the understanding for the readers, as it is not clear many times if the work presented is new or they come from a previous published work. Authors should be clearer about that. Some conclusions done here are overstatements in agreement with the figure that are presented. Some figures would need a better explanation of the methodology, other would require some controls, but perhaps they are not overstatements if they come from other works with extra information that support these statements. At this point, it is difficult to me to judge this, because the lack of references and the lack of this information clearly described on the text. For examples, see the pdf document.
- In general, a better conductive line for the entire manuscript would be of interest for the reader. The topics appear suddenly without connection and sometimes without a proper introduction.
- Figures are all with formatting problems. The labelling are not properly done. Perhaps is a pdf conversion problem, but a revision of all figure should be done.
- More specific comments are presented in the attached pdf.

Author Response
Manuscript ID: jof-1027827
Title: Physiological basis of smut infectivity in early stages of sugar cane colonization
Journal of Fungi
Reviewers' comments AND ANSWERS (please see the pdf to locate all changes)
Reviewer #1: * Thus, summarize and put in perspective the knowledge on this important plant pathogen is of interest. However, the review falls short in that it does not really cover the full knowledge on pathogenic mechanisms for this fungus and neither they are immersed in the general knowledge of other plant pathogenic fungi. Due to the reduced knowledge on this fungus, I consider important to cover the whole data available and compare it with the knowledge on other pathogenic fungi in a more systematic way. Authors focus on two mechanisms, quorum sensing and cytoskeletal component rearrangements. Although these two aspects of infection are of interest, general effector biology is currently a field of great interest and several genomic, transcriptomic and proteomic works available for S. scitamineum highlight the presence of virulence factors with effect on different biology process (Que et al. 2014; Su et al. 2015; Dutheil et al. 2016; Barnabas et al. 2017). In addition, the filamentation and mating processes are essential process for infection and several works describing regulators of this processes exists in this fungus. Thus, I consider that the systematic review of the infection mechanisms present in S. scitamineum rather than focusing in just two selected ones would be of a better interest.
ANSWER: We totally agree, so a lot of references have been included. Many of these references are detailed below, where the changes incorporated into the pdf are specified one by one. However, QS signal, the reorganization of the cytoskeleton and the germination are still considered main events in the first stages of infection to trigger another processes.
*A general problem I have found along the manuscript is the lack of references in important parts of the text. Frequently, this lack of references make difficult the review process, and it will make difficult the understanding for the readers, as it is not clear many times if the work presented is new or they come from a previous published work. Authors should be clearer about that.
ANSWER: We fully agree, so this has been thoroughly corrected. Many of the references included are detailed below, where the changes incorporated into the pdf are specified one by one.
*Some conclusions done here are overstatements in agreement with the figure that are presented. Some figures would need a better explanation of the methodology, other would require some controls, but perhaps they are not overstatements if they come from other works with extra information that support these statements. At this point, it is difficult to me to judge this, because the lack of references and the lack of this information clearly described on the text. For examples, see the pdf document.
ANSWER: Indeed, in many cases it was a lack of an adequate reference. It has been corrected.
*In general, a better conductive line for the entire manuscript would be of interest for the reader. The topics appear suddenly without connection and sometimes without a proper introduction.
ANSWER: It has been tried to achieve by looking for the connection between sections as can be seen between sections 2 and 3 (263-264 lines), 4 and 5 (439-445 lines) or 6 and (611-614 lines)
*Figures are all with formatting problems. The labelling are not properly done. Perhaps is a pdf conversion problem, but a revision of all figure should be done.
ANSWER: Indeed, it was a problem for the conversion to pdf. I hope it is already solved.
*More specific comments are presented in the attached pdf (some changes to the pdf are specified below)
a) The life cycle of the Ustilaginales point appears now secondly, as it was suggested. The title in now: 2. How is the life cycle of the Ustilaginales like Sporisorium scitamineum? (line 70). In this part, some references have been included. Fig. 2 (it was Fig. 3 previously) has been modified in order to incorporate S. scitamineum important specificities of life cycle.
b) Factors of infectivity title has been replaced by 3.A quorum signal (QS) triggers the infection as it was suggested (line 266).
c)A quorum signal (QS) triggers the infection point (267-344 lines) has been thoroughly checked. Many references about QS molecules have been included:
Moleleki L.N.; Pretorius R.G.; Tanui C.K.; Mosina G.; Theron J. A quorum sensing-defective mutant of Pectobacterium carotovorum brasiliense 1692 is attenuated in virulence and unable to occlude xylem tissue of susceptible potato plant stems. Mol. Plant Pathol. 2017, 18 (1), 32-44.
Hogan, D.A. Talking to themselves: Autoregulation and quorum sensing in fungi. Eukaryotic Cell 2006, 5: 613–619.
Barriuso J.; Hogan D.A.; Keshavarz T.; Martínez M.J. Role of quorum sensing and chemical communication in fungal biotechnology and pathogenesis. FEMS Microbiol Rev. 2018, 42(5), 627-638.
Mehmood A.; Liu G.; Wang X.; Meng G.; Wang C.; Liu Y. Fungal Quorum-Sensing Molecules and Inhibitors with Potential Antifungal Activity: A Review. Molecules 2019, 24(10):1950.
Hornby, J.M.; Jensen, E.C.; Lisec, A.D.; Tasto, J.J.; Jahnke, B.; Shoemaker, R.; Dussault, P.; Nickerson, K.W. Quorum sensing in the dimorphic fungus Candida albicans is mediated by farnesol. Appl. Environ. Microbiol. 2001, 67, 2982–2992.
Pathak P., Sahu P. Perspective of Quorum Sensing Mechanism in Candida albicans. In: Pallaval Veera Bramhachari (eds) Implication of Quorum Sensing System in Biofilm Formation and Virulence, 2018. Springer, Singapore.
Macko, V., Staples, R.C.; Gershon, H.; Renwick, J.A. Self-inhibitor of bean rust uredospores: methyl 3,4-dimethoxycinnamate. Science 1970, 170:539-540.
Lingappa, B.T.; Lingappa. Y. Role of auto-inhibitors on mycelial growth and dimorphism of Glomerella cingulata. Gen. Microbiol. 1969, 56:35-45.
Yajima, A. Recent progress in the chemistry and chemical biology of microbial signaling molecules: quorum-sensing pheromones and microbial hormones. Tetrahedron Lett. 2014, 55, 27732773ne.
Vitale, S.; Di Pietro, A.; Turrà, D. Autocrine pheromone signalling regulates community behaviour in the fungal pathogen Fusarium oxysporum. Microbiol. 2019, 4, 1443–1449.
d) 2 (now, Fig. 3) has been modified and explained so that it is better understood (322-323 lines).
e) Pathogenicity factors section has been thoroughly checked. Many references about virulence factors have been incorporated (including all of them suggested by the referee):
Barnabas, L.; Ashwin, N.M.R.; Kaverinathan, K.; Trentin, A.R.; Pivato, M.; Sundar, A. R.; Malathi, P.; Viswanathan, R.; Carletti, P.; Arrigoni, G.; Masi, A.; Agrawal, G.K.; Rakwal, R. In vitro secretomic analysis identifies putative pathogenicity-related proteins of Sporisorium scitamineum–The sugarcane smut fungus. Fungal Biol. 2017, 121(3), 199-211.
Teixeira-Silva, N.S.; Schaker, P.D.C.; Rody, H.V.S.; Maia, T.; Garner, C.M.; Gassmann, W.; Monteiro-Vitorello, C.B. Leaping into the Unknown World of Sporisorium scitamineum Candidate Effectors. Fungi 2020, 6, 339.
Dutheil, J.Y.; Mannhaupt, G.; Schweizer, G.; Sieber, C.M.K.; Münsterkötter, M.; Güldener, U.; Schirawski, J.; Kahmann, R.A Tale of genome compartmentalization: the evolution of virulence clusters in smut fungi. Genome Biol. Evol. 2016, 8(3), 681–704.
Su, Y.; Xu, L.; Wang, S.; Wang, Z.; Yang, Y.; Chen, Y.; Que, Y. Identification, phylogeny, and transcript of chitinase family genes in sugarcane. Nat. Sci. Rep. 2015, 5, 10708.
Su, Y.; Guo, J.; Ling, H.; Chen, S.; Wang, S.; Xu, L.; Allan, A.C.; Que, Y. Isolation of a novel peroxisomal catalase gene from Sugarcane, which is responsive to biotic and abiotic stresses. PLoS One 2014, 9 (1), e84426.
Sánchez-Elordi E.; de Los Ríos L.M.; Vicente C.; Legaz M.E. Polyamines levels increase in smut teliospores after contact with sugarcane glycoproteins as a plant defensive mechanism. J Plant Res. 2019, 132(3):405-417.
Valdés-Santiago L.; Guzmán-de-Peña D.; Ruiz-Herrera J. Life without putrescine: disruption of the gene-encoding polyamine oxidase in Ustilago maydis odc mutants. FEMS Yeast Res. 2010, 10(7):928-40.
Guevara-Olvera L.; Xoconostle-Cázares B.; Ruiz-Herrera J. Cloning and disruption of the ornithine decarboxylase gene of Ustilago maydis: evidence for a role of polyamines in its dimorphic transition. Microbiology, 1997, 143:2237–2245.
Chang, C., Cai, E., Deng, Y. Z., Mei, D., Qiu, S., Chen, B., Zhang, L.; Jiang, Z. cAMP/PKA signalling pathway regulates redox homeostasis essential for Sporisorium scitamineum mating/filamentation and virulence. microbiol. 2019, 21(3), 959-971.
Liu, C.; Atanasov, K.E.; Arafaty, N.; Murillo, E.; Tiburcio, A.F.; Zeier, J.; Alcázar, R.. Putrescine elicits ROS‐dependent activation of the salicylic acid pathway in Arabidopsis thaliana. Plant Cell Environ, 2020, 43(11), 2755-2768.
Sánchez-Elordi, E.; Baluška, F.; Echevarría, C.; Vicente, C.; Legaz, M.E. Defence sugarcane glycoproteins disorganize microtubules and prevent nuclear polarization and germination of Sporisorium scitamineum teliospores. Plant Physiol. 2016, 200, 111-123.
Sánchez-Elordi, E.; Baluska, F.; Vicente, C.; Legaz, M.E. Sugarcane glycoproteins control dynamics of cytoskeleton during teliospore germination of Sporisorium scitamineum. Progr. 2019, 18, 1121-1134.
f) Connection between sections 4 and 5 (439-445 lines) has been written.
g) In section 5 required references have been incorporated and the paragraphs referring to the organization of actin/MTs have been related to previous studies so that it is better understood (480-522 lines). Fig. 7 about nuclear migration (Fig. 11 before) is now connected with this section and referenced (524-531 lines).
Vasquez, R.J.; Howell, B.; Yvon, A.M.C.; Wadsworth, P.; Cassimeris, L.; Nanomolar concentrations of nocodazole alter microtubule dynamic instability in vivo and in vitro. Mol. Biol. Cell 1997, 8, 973–
Snyder, M.; Gehrung, S.; Page, B.D. Studies concerning the temporal andgenetic control of cell polarity in Saccharomyces cerevisiae. Cell Biol. 1991, 11,515–532
Pillai, M.C.; Baldwin, J.D.; Cherr, G.N. Early development in an algalgametophyte: role of the cytoskeleton in germination and nucleartranslocation. Protoplasma 1992, 170, 34–45.
Åström, H.; Sorri, O.; Raudaskoski, M.; Role of microtubules in the movementof the vegetative nucleus and generative cell in tobacco pollen tubes. PlantReprod. 1995, 8, 61–69.
Sánchez-Elordi, E.; Baluska, F.; Vicente, C.; Legaz, M.E. Sugarcane glycoproteins control dynamics of cytoskeleton during teliospore germination of Sporisorium scitamineum. Progr. 2019, 18, 1121-1134.
h) Similarly, references required by the referee in sections 6 and 7 have been incorporated.
Sánchez-Elordi, E.; Vicente-Manzanares, M.; Díaz, E.; Legaz, M.E.; Vicente, C. Plant–pathogen interactions: Sugarcane glycoproteins induce chemotaxis of smut teliospores by cyclic contraction and relaxation of the cytoskeleton. South African J. Bot. 2016, 105, 66–78.
Sánchez-Elordi, E.; Baluška, F.; Echevarría, C.; Vicente, C.; Legaz, M.E. Defence sugarcane glycoproteins disorganize microtubules and prevent nuclear polarization and germination of Sporisorium scitamineum teliospores. Plant Physiol. 2016, 200, 111-123.
Sánchez-Elordi, E.; Baluska, F.; Vicente, C.; Legaz, M.E. Sugarcane glycoproteins control dynamics of cytoskeleton during teliospore germination of Sporisorium scitamineum. Progr. 2019, 18, 1121-1134.
i) Cytoagglutination assay explication has been included (726-772 lines)
The referees' suggestions have been incorporated, as can be seen in the manuscript. Mostly, the problem was the lack of references. Other minor errors have been corrected in the manuscript (commas, repetitions ..)
Reviewer 2 Report
In this review, the author explains the physiological basis of sugarcane smut infectivity. They have explained the factors of infectivity, the life cycle of the Ustilaginals, pathogenicity factors, the role of the cytoskeleton in teliospore germination, the role of the cytoskeleton in teliospore motility, and the GTPases as mediators of the signal of organization of the cytoskeleton. The manuscript is very well written to have some grammatical errors (Part 2 2nd paragraph change pathogen over time on that to pathogen overtime on that).
I have found that the author use already published images of their own article. This is not ethical. Please use unpublished images.
Figure 7 of this paper already published in South African Journal of Botany "Plant–pathogen interactions: Sugarcane glycoproteins induce chemotaxis of smut teliospores by cyclic contraction and relaxation of the cytoskeleton". Figure 4 has the same image as Figure 7 of this article. Please use unpublished images in this article.
Author Response
Manuscript ID: jof-1027827
Title: Physiological basis of smut infectivity in early stages of sugar cane colonization
Journal of Fungi
Reviewers' comments AND ANSWERS (please see the pdf to locate all changes)
Reviewer #2: * The manuscript is very well written to have some grammatical errors (Part 2 2nd paragraph change pathogen over time on that to pathogen overtime on that).
ANSWER: It was corrected
*I have found that the author use already published images of their own article. This is not ethical. Please use unpublished images. Figure 7 of this paper already published in South African Journal of Botany "Plant–pathogen interactions: Sugarcane glycoproteins induce chemotaxis of smut teliospores by cyclic contraction and relaxation of the cytoskeleton". Figure 4 has the same image as Figure 7 of this article. Please use unpublished images in this article.
ANSWER: This image has been replaced by another obtained by TEM where teliospore invagination can be observed (Fig. 8, line 600).
The referees' suggestions have been incorporated, as can be seen in the manuscript. Mostly, the problem was the lack of references. Other minor errors have been corrected in the manuscript (commas, repetitions ..)
Round 2
Reviewer 1 Report
Authors have satisfactory made all requested changes and manuscript is in a good shape for publication.
Reviewer 2 Report
I am happy with the author's reply. MS can be accepted as it is.